# Processing and Properties of Sintered W/Steel Composites for the First Wall of Future Fusion Reactor

**Vishnu Ganesh** [1,*], **Daniel Dorow-Gerspach** [1,*], **Martin Bram** [1,2], **Jan Willem Coenen** [1], **Marius Wirtz** [1], **Gerald Pintsuk** [1], **Werner Theisen** [2] and **Christian Linsmeier** [1]

1 Forschungszentrum Jülich GmbH, Institut für Energie- und Klimaforschung, 52425 Jülich, Germany
2 Institut für Werkstoffe, Lehrstuhl Werkstofftechnik, Ruhr-Universität Bochum, 44801 Bochum, Germany
* Correspondence: v.ganesh@fz-juelich.de (V.G.); d.dorow-gerspach@fz-juelich.de (D.D.-G.)

**Abstract:** Functionally graded tungsten/steel composites are attractive to be used as an interlayer to join tungsten (W) and steel for the first wall of future fusion reactor to reduce the thermally induced stresses arising from the different coefficient of thermal expansion (CTE) of W and steel. W/steel composites, with three W contents: 25, 50 and 75 vol% W, will serve as individual sublayers of this functionally graded material. Therefore, the present work exploits an emerging sintering technique, field-assisted sintering technology, to produce these composites. Firstly, a systematic parameter study was conducted aiming to reduce the residual porosity to a minimum while keeping the formation of intermetallic phases at the W/steel interface at a low level. The optimized composites 25, 50 and 75 vol% W achieved a relative density of 99%, 99% and 96%, respectively. Secondly, mechanical tests at elevated temperatures reveal that these composites are ductile above 300 °C, which is the minimum operating temperature of the first wall. Lastly, the measured CTE, specific heat capacity and thermal conductivity were consistent with the theoretically expected values.

**Keywords:** W/steel composites; FAST/SPS; first wall; thermal analysis; mechanical analysis





## 1. Introduction

In the first wall of a future fusion reactor, the plasma-facing side will be made of tungsten (W) and the structural material (below it) will be made of steel. This demands the joining of W and steel; however, the direct joining of W and steel is challenging due to the mismatch in their coefficient of thermal expansion (CTE), which generates thermal stress peaks at their interface. A promising solution would be to introduce a functionally graded material (FGM), made of W/steel composites with varying volume concentration of W, as an interlayer to gradually change the CTE. This concept of using FGM as an interlayer with varying the volume concentration of W to mitigate the stresses has been already described in several previous works [1–3]. According to numerical calculations, FGM consisting of three sublayers (25, 50, 75 vol% W) is a suitable compromise with regard to the manufacturing effort and stress reduction [1]. For simplicity, the compositions are labelled as 25 W, 50 W and 75 W in this work.

There are several manufacturing routes to produce such kinds of FGM and these can be primarily categorized into magnetron sputtering, plasma spraying and powder metallurgy. Magnetron sputtering has the drawback of producing very thin FGM (~3 μm) and hence, it is not a suitable route [4]. Contrarily, plasma spraying is a suitable route to produce thick (1 to 2 mm) FGM. Recently, vacuum plasma spraying (VPS) was used to produce 1.2 mm thick FGM [5]. However, VPS requires the use of large vacuum chamber and such systems are expensive and not widely available [6]. Several studies have, instead, used atmospheric plasma spraying (APS). However, the produced composites have several drawbacks, such as: high porosity (10 to 15%) [4], high brittleness [4] and low thermal conductivity (15 to 20 W/m.K) due to the porous and lamellar structure [7]. Powder

metallurgy can produce homogenous and thick FGM layers with better properties. A conventional hot pressing technique was used to produce composites by using W and steel powders of different particle size fractions (PSF) [8]. At first, the powders were manually mixed and then sintered at 2000 °C, 6 GPa. Although the resulting composites were dense, this high sintering temperature resulted in the formation of high amount of thick $Fe_xW_y$ intermetallic compound (IMC) belt at the W–steel interfaces; e.g., in 50 W the thickness of this IMC belt was around 3 to 4 μm. IMCs are brittle and therefore, detrimental for mechanical properties. High amounts of IMC can even deteriorate the structural integrity of the composite in the case of thermal cycling and/or thermal shocks during the operation of the fusion reactor. Thus, the amount of IMC must be reduced to a minimum. Electrical current-activated/assisted sintering (ECAS) techniques are attractive to achieve this aim due to enabling reduction in sintering temperature and sintering time, and therefore limiting the formation of IMCs.

Below, a brief overview of developments in producing FGM by several ECAS techniques is given. A dense W/Fe FGM was achieved by resistance sintering under ultra-high pressure (9 GPa) and low sintering time of 60 s by first manually mixing the W and Fe powders of different PSF. Nevertheless, the resulting FGM contained a high amount of IMC; in the 50 W composite, a 2 to 5 μm thick IMC belt was formed [9]. Another attempt to limit the formation of IMC was the sintering of FGM in a lab-scale ultra-fast electro discharge sintering (EDS) machine. This technique makes it possible to sinter the powders within milliseconds [10]. W/Fe composites were produced starting from manually mixed W and Fe powders with different PSF. However, the produced composites were not dense. Therefore, energetic ball milling was used, at first to mix the W and Fe powders and then sintered using EDS [10]. Although the resulting sintered composites were dense, two drawbacks arose: Firstly, due to the ball milling, the composite had an interlamellar-type microstructure. This is due to the fact that, ball milling of ductile and brittle components (W is brittle and steel is ductile) leads to this interlamellar-type microstructure [11]. These fine lamellae may act as local stress concentration regions [12]. This can reduce the structural integrity of the composite. Secondly, ball milling increases the quantity of W–steel interfaces and as the IMC tend to form mainly at these interfaces; any increase in these interfacial areas should be avoided. For this reason, in another study, W/steel composites were sintered using EDS by omitting ball milling during powder processing. The W and steel powders with different and broad PSFs were manually mixed using a tumble mixer [12]. The resulting sintered composites had a high porosity (5% for 50 W and 20 to 50% for 75 W), resulting in poor mechanical properties. Additionally, due to its dynamic nature and densification within milliseconds EDS causes inhomogeneous consolidation and upscaling the technique for the manufacture of large plasma-facing components is questionable [12].

As an alternative, field-assisted sintering technology/spark plasma sintering (FAST/SPS) belongs to the ECAS techniques as well and has the capability to sinter homogenous large-scale compacts, and industrial-scale equipment are also available. FAST/SPS is an advanced sintering process that utilizes resistance heating and mechanical pressure to consolidate metallic or ceramic powders in a shorter time as compared to conventional powder metallurgy processing, thereby limiting the formation of IMC [13–15]. Up to now, only a few studies have been published describing the production of W/steel composites via FAST/SPS. Tan et al. [16] produced W/steel composites and FGM by mixing the W and steel powders via energetic ball milling for 5 h and 10 h, and sintered this with the sintering parameter of 1050 °C, 45 MPa, 5 min. However, a high amount of IMC was formed due to reasons mentioned before. In the 50 W composite, thick IMC phase formed at the W–steel interface. Its thickness increased from 5 μm for a milling time of 5 h to 8 μm for a milling time of 10 h. Therefore, Koller et al. [17] mixed the powders manually and omitted the ball milling to limit the amount of IMC formation at the W–steel interface. Sintering was carried out at 1100 °C, 80 MPa, 2 min. However, the resulting composites still had few drawbacks: Firstly, the resulting composites were not dense; the resulting 20 W, 43 W and 69 W composites had a relative density of roughly 98%, 95% and 82%.

Secondly, the sintering temperature of 1100 °C was still high enough to form a thick IMC belt at the W–steel interface; however, the exact thickness was not discussed in this work. Nevertheless, the high amount of IMC (20 W, 43 W, 69 W contained 21%, 12%, 13% IMC) indicates the processing conditions are not optimum. Thirdly, because of the use of fine (<20 μm), irregularly shaped W powder, the W particles form agglomerates and after the sintering, these agglomerates acts as weak spots in the microstructure due to weak W–W bonding, which is not surprising at a sintering temperature of 1100 °C. Appearance of these weak spots resulted in poor elastic modulus for the 43 W and 69 W composites.

To overcome these drawbacks, the following key points are considered in the present study: Firstly, spherical W and steel powders were selected, which do not form agglomerates. Secondly, the powder mixing of W and steel powder was carried out by manually mixing using a tumble mixer. High energy ball milling was consciously omitted considering the drawbacks mentioned before. Thirdly, the sintering temperature must be kept reasonably low to limit the amount of IMC. Fourthly, W has—due to its high melting point—a low sintering activity, which leads to weak W–W bonds at moderate sintering temperatures. To address this fact and to reduce the W–steel interface to a minimum, we decided to embed relatively large W particles in a matrix of smaller steel particles; in an ideal case, a percolating steel network is sintered around the W particles, which then is mainly responsible for the mechanical properties of the FGM.

Thus, in this work, W/steel composites with three different volume concentrations of W (25, 50 and 75 vol%) were manufactured. The aim of this work was to produce and characterize dense composites with the lowest amount of detrimental brittle IMC. Therefore, firstly, the influence of sintering parameters, sample thickness and PSF of the starting powders on density and possible IMC formation were investigated. Secondly, optimum processing parameters were discovered and optimized composites were manufactured. Finally, mechanical and thermophysical analysis of these optimized composites were performed considering its application as a FGM interlayer.

## 2. Materials and Methods

### 2.1. Powder Preparation

Spherical W and steel powders were used as the starting materials. Two batches of W powders were purchased, one from China Tungsten Online (Xiamen) Manu. & Sales Corp, (Fujian, China), and the other one from Tekna Advanced Materials, Québec, Canada. The PSFs of the powders were $+10/-30$ μm and $+30/-60$ μm, respectively. The $D_{50}$ were ~17 μm and ~50 μm, respectively. $D_{50}$ is the value of particle size at 50% in the cumulative particle size distribution and represents the median particle size of a powder. Similarly, two batches of steel powders were purchased from Nanoval GmbH & Co. KG, Berlin, Germany. The first batch had a PSF of $+10/-20$ μm and $D_{50}$ of 13 μm; the second batch had a PSF of $+3/-13$ μm and $D_{50}$ of 7 μm. The elemental composition of the steel powders was similar to that of Eurofer 97, a reduced activation martensitic/ferritic steel [18–20]. The powder handling was performed in a glove box with an inert atmosphere to prevent the oxidation of powders and the mixing of the powders was carried out as follows: the respective powders of desired PSF were weighed accordingly and filled in a plastic container, which was then sealed (inside the glove box) to maintain an inert atmosphere. Then, this container was removed from the glove box, mounted in a tumble mixer and mixed for 72 h to obtain homogenously mixed powders. Scanning electron microscopy (SEM) images of some of the mixed powders are given as Figure S1 in Supplementary Materials.

### 2.2. Sintering Methodology

The sintering was performed using lab-scale FAST/SPS equipment (HP D-5 from FCT Systeme GmbH) using graphite tools (die and punches). The diameter of the die and punches was 20 mm. The tools were fabricated from an isostatically pressed graphite material (ISOSTATIC 2334 from MERSEN, Rhone, France) with a compressive strength of

230 MPa. The geometry of the punch was optimized to withstand a pressure of 125 MPa, whose technical drawing is provided as Figure S2 in Supplementary Materials.

The mixed powder was filled in this die and then pre-pressed to 125 MPa pressure. The powder and the die were separated by a 0.025 mm thick molybdenum foil to reduce the diffusion of carbon from the graphite tools to the composite material [21]. The sintering was performed under mild vacuum (~0.1 mbar). The pulse ON/OFF times of the pulsed DC current for the FAST/SPS cycle were 25 ms/5 ms, respectively. The temperature was monitored and controlled using a vertical pyrometer focused on the bottom of a bore drilled in the upper punch. The heating rate was 100 K/min. The temperature was kept constant after reaching the desired sintering temperature for the desired amount of dwell time (sintering time). At the end of the sintering time, the current was switched off, enabling the sample to cool down rapidly.

Three compositions with 25 vol% W, 50 vol% W and 75 vol% W were sintered as summarized in Table 1. PSF and sintering parameters (temperature, time and pressure) were optimized to achieve high densification. Moreover, the 25 W was sintered varying the sample thickness in order to study its effect. The cross-section of the sintered composite was investigated using SEM images; the residual porosity and IMC were determined by image analysis. Several SEM micrographs were taken at different magnifications to show specific microstructural details appearing on the meso- and micro-scale; this was carried out so as to include the micro- as well as meso-scale porosities/voids for the residual porosity measurement. The mean value of this represents here the amount of porosity of the sintered composite and its standard deviation is represented as an error band. Similarly, the amount of IMC was also determined from image analysis. This porosity and the amount of IMC was helpful to determine the optimized sintering parameters.

**Table 1.** PSF of the W and steel powders for different compositions.

| Composition | Nomenclature | PSF of W | PSF of Steel |
|:---:|:---:|:---:|:---:|
| 25 W | $25W_{10-30}+75S_{10-20}$ | $+10/-30$ μm | $+10/-20$ μm |
| 50 W | $50W_{10-30}+50S_{10-20}$ | $+10/-30$ μm | $+10/-20$ μm |
| | $50W_{10-30}+50S_{3-13}$ | $+10/-30$ μm | $+3/-13$ μm |
| 75 W | $75W_{30-60}+25S_{10-20}$ | $+30/-60$ μm | $+10/-20$ μm |

*2.3. Characterization of Composites*

After the optimization of sintering parameters, composites were freshly sintered applying these optimized parameters and further characterized. For the characterization, specimens were cut out of these freshly sintered composites using wire electric discharge machining to appropriate dimensions required for the different characterization methods.

2.3.1. Mechanical Characterization

Four-point bending tests were performed on specimens of 12 mm × 1 mm × 1 mm size at 20 °C, 100 °C, 300 °C and 550 °C (under vacuum). The flexural stress ($\sigma_f$) and strain ($\varepsilon_f$) were calculated based on ASTM D7265/D7264M [22]. For correct interpretation of the results, it must be noted that the formula used for stress and strain calculation is only valid in the elastic regime. Furthermore, there is no standard for performing bending tests on such small specimens, so these stress–strain curves must be read with care.

2.3.2. Thermophysical Characterization

The density and the relative density of the composites were measured using the Archimedes' principle using ethanol as fluid at room temperature (20 °C). Relative density is the ratio of actual density measured via Archimedes' principle to the theoretical density of a particular composition. The theoretical density of the composition represents the density of the composition, considering no porosity (means 100% dense composite) and is calculated by the rule of mixture in the present work. Thermophysical characterization was performed by dilatometer, dynamic differential scanning calorimetry (DSC), and laser

flash analysis (LFA). Sample geometries and testing temperatures are summarized in Table 2. These tests were performed to determine the following respective quantities: CTE, specific heat capacity ($c_p$) and thermal conductivity ($\lambda$). All the tests were performed under Ar atmosphere.

**Table 2.** Overview of thermophysical tests and testing conditions.

| Test | Equipment | Specimen (mm) | Temperature (°C) |
|---|---|---|---|
| Dilatometer | LV75 from LINSEIS | $4 \times 2 \times 15$ | 20 to 1000 |
| DSC | DSC 404 F3 from NETZSCH | Ø 5 × 1.5 | 20 to 1000 |
| LFA | LFA427 from NETZSCH | $10 \times 10 \times 1.5$ | 20, 200, 400, 600, 800, 1000 |

The measured quantities were compared to their theoretical expected values. The theoretical expected temperature ($T$) dependent secant CTE ($\alpha_{comp}$) and specific heat capacity ($c_{p,comp}$) for different volume concentration of W ($V_W$) were calculated according to Equation (1) and Equation (2), respectively [1–3,23]. $\alpha_W$ and $\alpha_{Eurofer97}$ are the secant CTEs, $\rho_W$ and $\rho_{Eurofer97}$ are the densities, $c_{p,W}$ and $c_{p,Eurofer97}$ are the specific heat capacities of pure W and Eurofer 97 steel, respectively [24–26].

$$\alpha_{comp}(V_W, T) = V_W \alpha_W(T) + (1 - V_W)\alpha_{Eurofer97}(T) \tag{1}$$

$$c_{p,comp}(V_W, T) = \frac{c_{p,W}(T)\rho_W V_W + c_{p,Eurofer97}(T)\rho_{Eurofer97}(1 - V_W)}{\rho_W V_W + \rho_{Eurofer97}(1 - V_W)} \tag{2}$$

Unlike CTE and $c_p$, there is no specific relation to predict the theoretical thermal conductivity of such composites, as it depends predominantly on the spatial microstructural arrangement of W and steel constituents. Therefore, simple upper ($\lambda_{upper-bound,comp}$) and lower bound ($\lambda_{lower-bound,comp}$) models were suggested here to compare the measured values (see Equations (3) and (4)) [1,4]. Here, $\lambda_W$ and $\lambda_{Eurofer97}$ represent the thermal conductivity of pure W and Eurofer 97 steel, respectively [24,25].

$$\lambda_{upper-bound,comp}(V_W, T) = V_W \lambda_W(T) + (1 - V_W)\lambda_{Eurofer97}(T) \tag{3}$$

$$\lambda_{lower-bound,comp}(V_W, T) = \left(\frac{1 - V_W}{\lambda_{Eurofer97}(T)} + \frac{V_W}{\lambda_W(T)}\right)^{-1} \tag{4}$$

## 3. Results and Discussion

### 3.1. Optimizing the Sintering Parameters for 25 W

Figure 1a shows the residual porosities of the 25 W composites sintered at three sintering temperatures for a sintering time of 5 min and at a pressure of 50 MPa. The thicknesses of the sintered composites were also varied to 0.75 mm, 2 mm and 3 mm. The increase in temperature results in the decrease in residual porosity. For the composite sintered at 900 °C, the majority of pores were present inside the steel matrix and with the increase in sintering temperature these pores completely closed. The residual porosity for composite sintered at 1100 °C reduced to less than 1%, but resulted in a high amount of IMC (~6%). These IMCs were not only present at the W–steel interfaces but also in steel regions close to W particles, mostly along the grain boundaries of the steel matrix. For thinner composites (2 mm and 0.75 mm), the increase in temperature from 1000 °C to 1100 °C did not have a significant effect on the residual porosity. For a sintering temperature of 1000 °C, the reduction in the consolidate's thickness from 3 mm to 2 mm clearly reduced the residual porosity to less than 1%. This reduction in porosity with the reduction in sample thickness is due to the wall friction effect which reduces the active powder compaction pressure for thicker samples. It is also worth mentioning that in these composites some rare occasional bigger pores (~2 to 3 μm) also appeared inside the steel matrix even

though the overall residual porosity was less than 1% (Please refer to Figures S4 and S5 in Supplementary Materials).

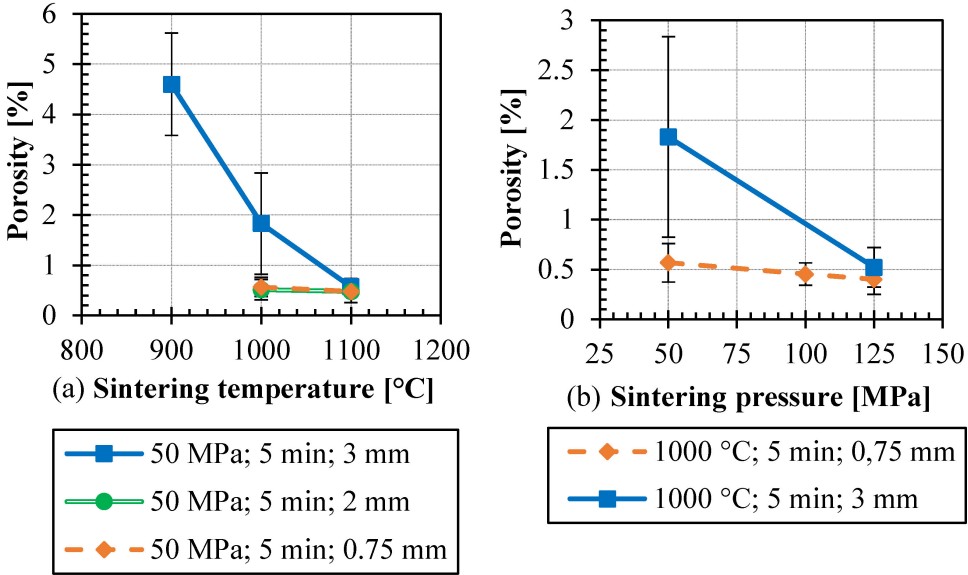

**Figure 1.** Residual porosity of sintered 25 W composites: (**a**) Effect of sintering temperature and consolidate's thickness; (**b**) Effect of sintering pressure and sample thickness.

In order to suppress the wall friction effect, the influence of sintering pressure was examined. For this, the sintering was also performed at higher pressure (100 and 125 MPa) as shown in Figure 1b. For the thinner sample (0.75 mm), the sintering pressure has no significant influence; it is because, as mentioned above, the wall friction effect comes into play only for thicker consolidates. For the thicker sample (3 mm), the pressure significantly improves the consolidation by counteracting the wall friction effect. Based on these results, the optimum sintering parameter for 25 W composite was found to be 1000 °C, 125 MPa, and 5 min and the cross-section of an optimized sintered 25 W composite is shown in Figure 2. Additional SEM micrographs are provided in Figures S3–S5 in Supplementary Materials.

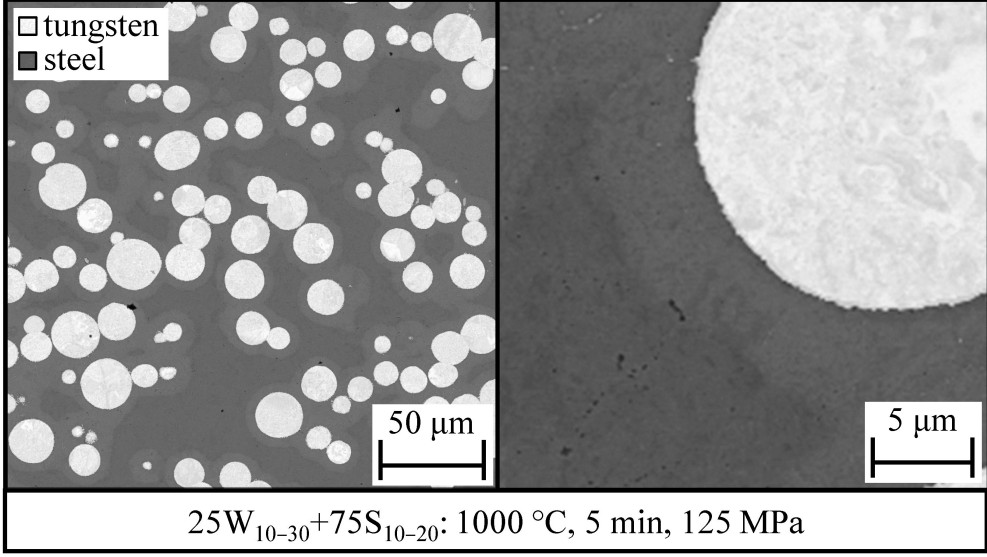

**Figure 2.** Cross-sectional SEM micrograph of 25 W composite sintered with optimized parameters (1000 °C, 125 MPa, 5 min).

### 3.2. Optimizing the Sintering Parameters for 50 W

At first, the composition $50W_{10-30}+50S_{10-20}$ was sintered at temperatures between 900 °C and 1100 °C at 50 MPa for time of 5 min. Also, based on the results of 25 W composites only two consolidates' thicknesses were considered (3 mm and 0.75 mm). Here as well, the increase in the sintering temperature resulted in better densification, as seen in Figure 3a. However, the porosity for the 3 mm thick composite sintered at 1000 °C was still high (7%). Increasing the temperature to 1100 °C reduced the porosity, but resulted in a higher amount of IMC (~14.87%). For the sintering temperature of 1000 °C, even the decrease in the composite's thickness from 3 mm to 0.75 mm did not significantly reduce the residual porosity. This implied that increasing the pressure would not lead to any sort of improvement and one needs to change the composition (PSF) in order to produce a dense composite.

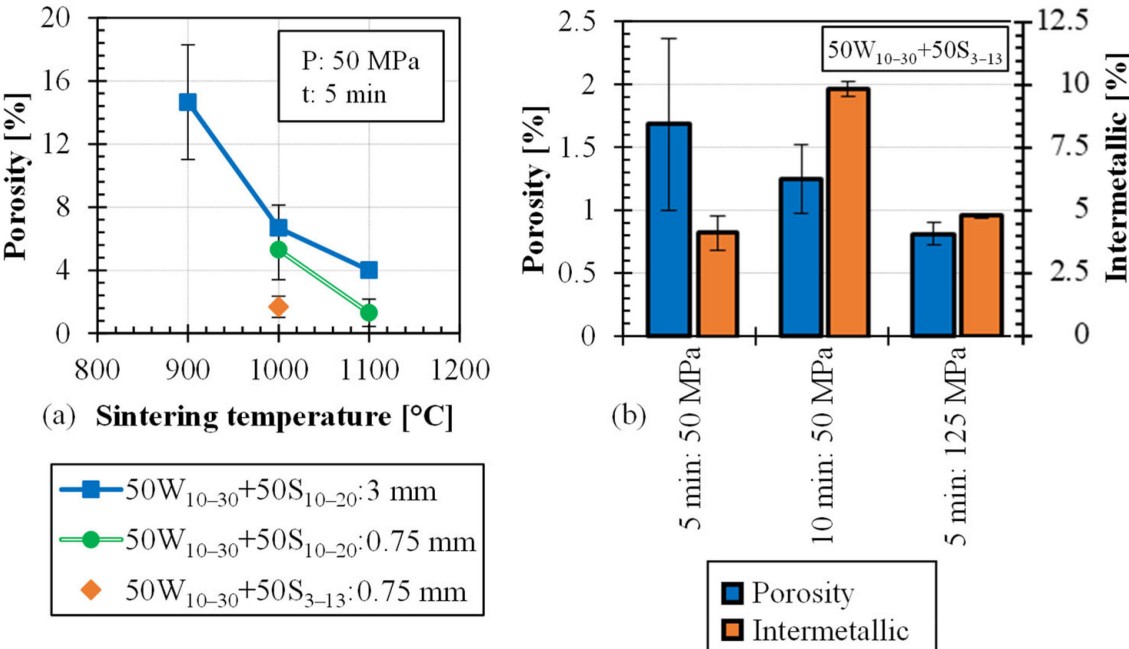

**Figure 3.** Residual porosity of sintered 50 W composites: (**a**) Effect of sintering temperature and consolidate's thickness as well as the effect of 50 W composition ($50W_{10-30}+50S_{10-20}$ and $50W_{10-30}+50S_{3-13}$); (**b**) Effect of sintering time and pressure for the 50 W composition ($50W_{10-30}+50S_{3-13}$).

Therefore, in order to achieve a higher density at a given sintering temperature of 1000 °C (the same as that of the optimized parameter for 25 W), another composition with finer steel powder, $50W_{10-30}+50S_{3-13}$, was investigated. In this case, a residual porosity of 1.6% was already achieved at 1000 °C, 5 min, 50 MPa, for a sample thickness of 0.75 mm, as seen in Figure 3a. Doubling the sintering time to 10 min led to a minor reduction in porosity, but the amount of IMC increased from 4.12% to 9.86%, as seen in Figure 3b. When the pressure was increased to 125 MPa, the residual porosity reduced to less than 1%, while the amount of IMC remained almost the same. Therefore, the optimized parameters for sintering the 50 W composite were found to be 1000 °C; 125 MPa; 5 min, and using the $50W_{10-30}+50S_{3-13}$ composition and the corresponding optimized composite is shown in Figure 4. Additional SEM micrographs are provided in Figure S6 in Supplementary Materials.

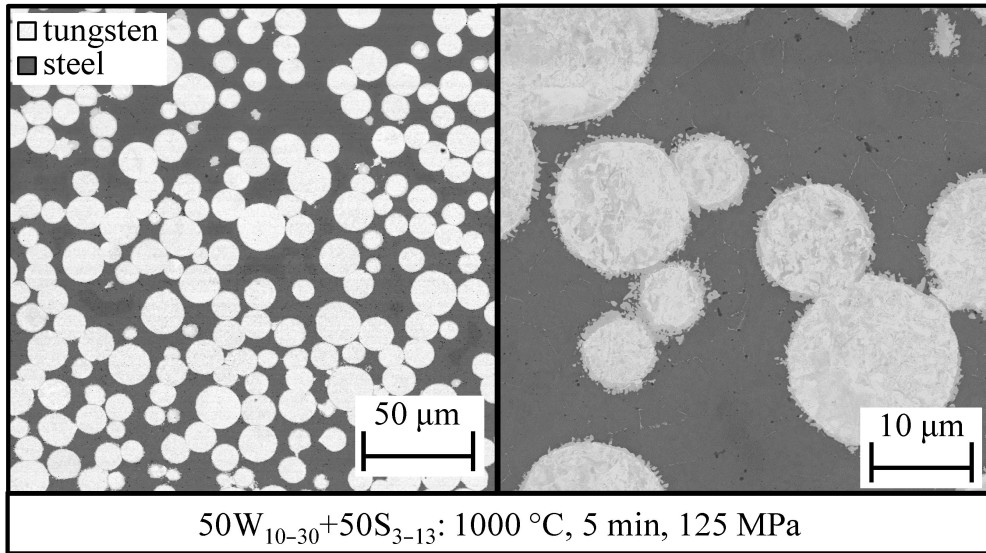

**Figure 4.** Cross-sectional SEM micrograph of 50 W composite sintered with optimized parameters (1000 °C, 125 MPa, 5 min).

### 3.3. Optimizing the Sintering for 75 W

The 75 W composition contains 75 vol% W and this makes the sintering process challenging because of two reasons: Firstly, the presence of a higher amount of W particles results in more W–W contact points, and generally, the metallurgical bonding of W–W particles is difficult. Secondly, usually a higher sintering temperature in the range of 1800 °C to 2000 °C is required to sinter pure W using a FAST/SPS process [27]. However, at such a high temperature the steel particles would melt. Contrarily, lowering the temperature would result in a higher number of unbonded W–W particles. Previous studies on the sintering of W/steel composite via a similar electric field-assisted sintering process revealed that for higher volume concentrations of W, the sintered composites contain higher amounts of unbonded W–W particles [12]. Hence, considering all these challenges, a coarser W powder of size +30/−60 μm instead of +10/−30 μm was used to reduce the amount of W–W particle contacts.

Thus, the resulting composition $75W_{30–60}+25S_{10–20}$ was sintered at various sintering temperature between 1000 °C and 1400 °C for a sintering time of 5 min and a pressure of 125 MPa. As seen in Figure 5, as the sintering temperature increases, the porosity decreases, but at the same time the amount of IMC also increases. It is not trivial to determine the optimized sintering parameters for 75 W composite: Firstly, at lower sintering temperature the porosity is higher, but at higher sintering temperatures, the amount of IMC becomes higher as well. Secondly, the sintering parameters for the 75 W composite should be the same as for 25 W and 50 W in order to produce the whole FGM in one step. Therefore, the optimized parameter set for 75 W was also considered to be 1000 °C, 5 min at 125 MPa, and the microstructure of the corresponding composite is shown in Figure 6. The left-hand side SEM micrograph shows that the composite is dense enough but, in some regions (as seen in the right-hand side SEM), the composite has higher porosity. Also, some of the W–W particles do not form any metallurgical bonding as represented by the red mark. Nevertheless, this composite will serve as the topmost 75 W layer of the FGM. Additional SEM micrographs are provided in Figure S7 in Supplementary Materials.

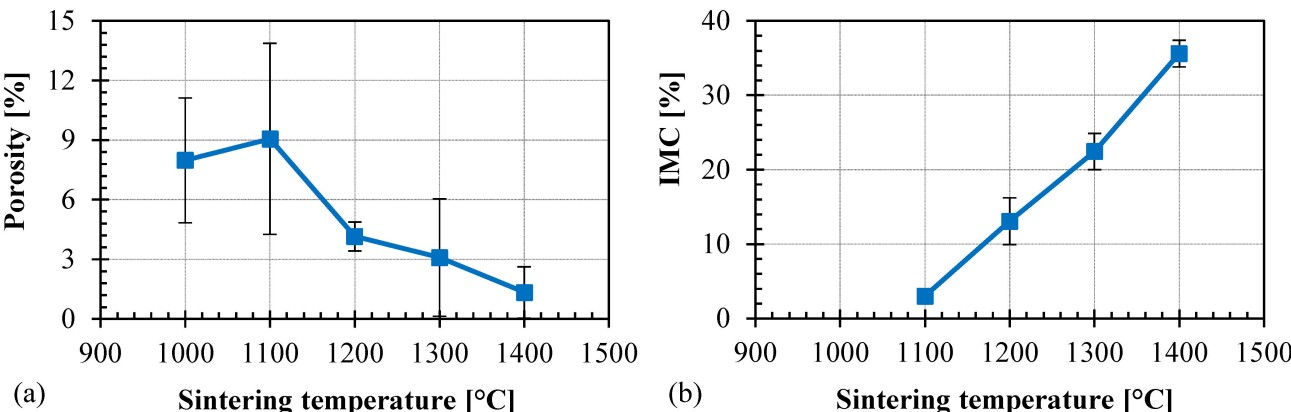

**Figure 5.** (**a**) Porosities and (**b**) IMC of 75 W composites sintered at different temperatures. (Note: for the sintering performed at 1000 °C, the amount of IMC could not be determined using image analysis as it was very minimal. However, it must be noted that even at 1000 °C, there will be generation of IMC, but its amount is not high enough to be captured by image analysis).

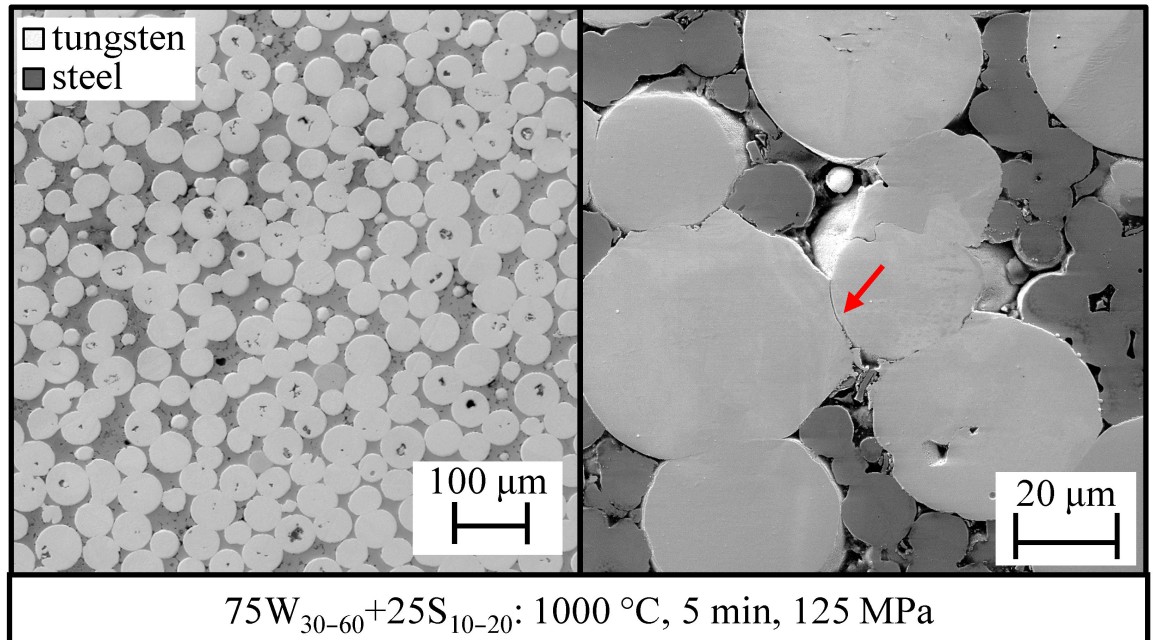

**Figure 6.** Cross-sectional SEM micrograph of 75 W composite sintered with optimized parameters (1000 °C, 125 MPa, 5 min).

*3.4. Properties of the Composites*

3.4.1. Comprehensive Interface Analysis

Conventional metallographic preparation by grinding and polishing did not enable microstructural details to be shown in sufficient magnification; therefore, in order to obtain a perfectly polished surface, focused ion beam cuts were made. The corresponding SEM micrographs (Figure 7) show individual phases, which are formed at the W/steel interface, along with their elemental composition. The composition was determined by energy-dispersive X-ray spectroscopy (EDX). The EDX results are summarized in Table 3. The main conclusions from the microstructure analysis are:

1.  Firstly, in all composites, a thin IMC phase with the composition $Fe_xW_yCr_z$ forms at the W–steel interface as confirmed by the EDX analysis (EDX-5). Nevertheless, the values of EDX-5 should be read with care since the thickness of this compound is lower

than the excitation area of EDX. The thickness of this IMC belt was roughly estimated to be 100 nm for 25 W, 200 nm for 50 W and 300 to 900 nm for 75 W composites.

2.  Secondly, nano-scale voids are present inside the steel matrix.

3.  Thirdly, in the 25 W composite (Figure 7a), a ferritic ($\alpha$) phase appeared in the steel matrix close to the W particle. This phase was formed because of the diffusion of W from the W particle to the steel matrix, resulting in around 8.9 wt% W inside this region, as listed by EDX-3 in Table 3. As W is a ferrite stabilizer, in this region no martensitic phase was formed during cooling. Instead, a ferritic phase formed. In the case of 50 W composite (Figure 7b), most of the steel phase became ferritic because of this interdiffusion of W (EDX-4). In the case of the 75 W composite as well (Figure 7c), most of the steel matrix was found to be ferritic.

4.  Fourthly, in the 25 W composite, the steel matrix further away from the W particle retains its original chemical composition, as indicated by EDX-2. This elemental composition is same to that of Eurofer 97 steel, which is a martensitic steel [19]. The SEM micrograph is this region clearly shows a martensitic phase structure. This observation was further confirmed by the cooling rate during the sintering process. From the time/temperature profile of the FAST/SPS cycle, the cooling rate between 1000 °C and 400 °C was estimated to be 210 K/min, which is significantly higher than the critical cooling rate (~5 K/min) to accomplish martensitic transformation [19,28].

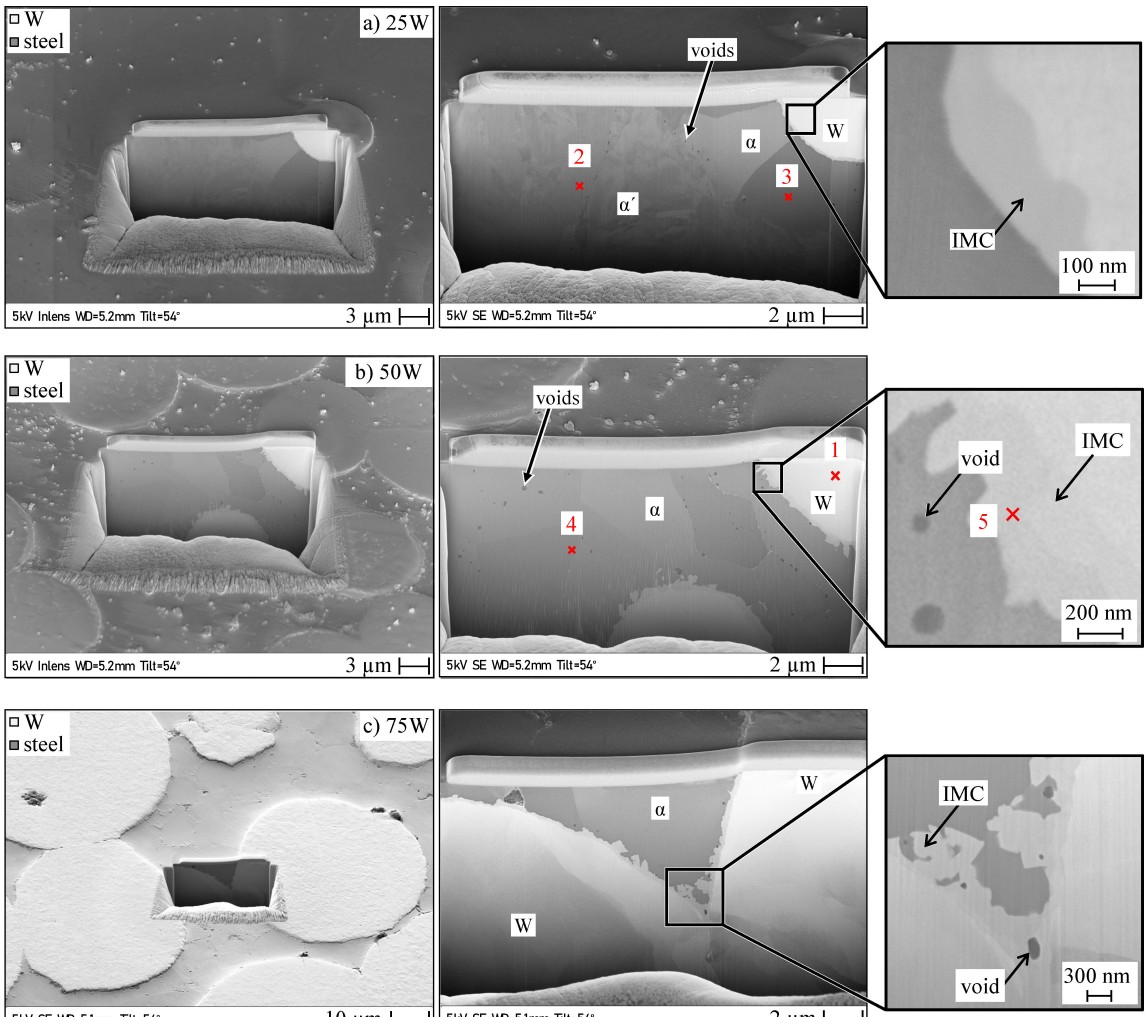

**Figure 7.** Comprehensive SEM micrographs showing individual phases: (**a**) 25 W; (**b**) 50 W; (**c**) 75 W.

**Table 3.** EDX spectrum analysis at different locations marked in Figure 7.

| EDX Spectrum (wt%) | Fe | Cr | W | V | Mn | Ta |
|---|---|---|---|---|---|---|
| EDX-1 | 0.5 | - | 99.5 | - | - | - |
| EDX-2 | 88.8 | 8.9 | 1.6 | 0.3 | 0.3 | 0.1 |
| EDX-3 | 82.8 | 8.3 | 8.3 | 0.2 | 0.4 | - |
| EDX-4 | 82.3 | 7.9 | 9.4 | 0.2 | 0.2 | - |
| EDX-5 | 31.9 | 4.3 | 63.6 | 0.1 | 0.1 | - |

Note: It should be noted that EDX analysis was not performed on the FIB cuts themselves, but they were performed on regions away from the FIB cuts. The marks in Figure 7 are intended to clarify the phases on which the measurements were made. Also, carbon is set as a deconvolution element during the EDX analysis.

### 3.4.2. Mechanical Properties

The flexural stress–strain curves of the composites are shown in Figure 8. The 25 W specimens tested at 300 °C and 550 °C did not fracture, but the test had to be stopped at around 8% flexural strain since this was the upper limit of the testing setup. The 50 W and 75 W composites were less ductile and, at 300 °C, failed at around 3% and 1% flexural strain, respectively. Above 550 °C, both 50 W and 75 W showed excellent ductility and failed at around 4% flexural strain. Moreover, the maximum flexural strength of 50 W and 75 W is significantly lower than that of 25 W. The strength of 75 W was slightly higher than that of 50 W, but still comparatively less than that of 25 W. This is because of the weak bonding between W particles. This implies that increasing the amount of W–W bonds in the W/steel composites is detrimental for the mechanical properties, resulting in early failure.

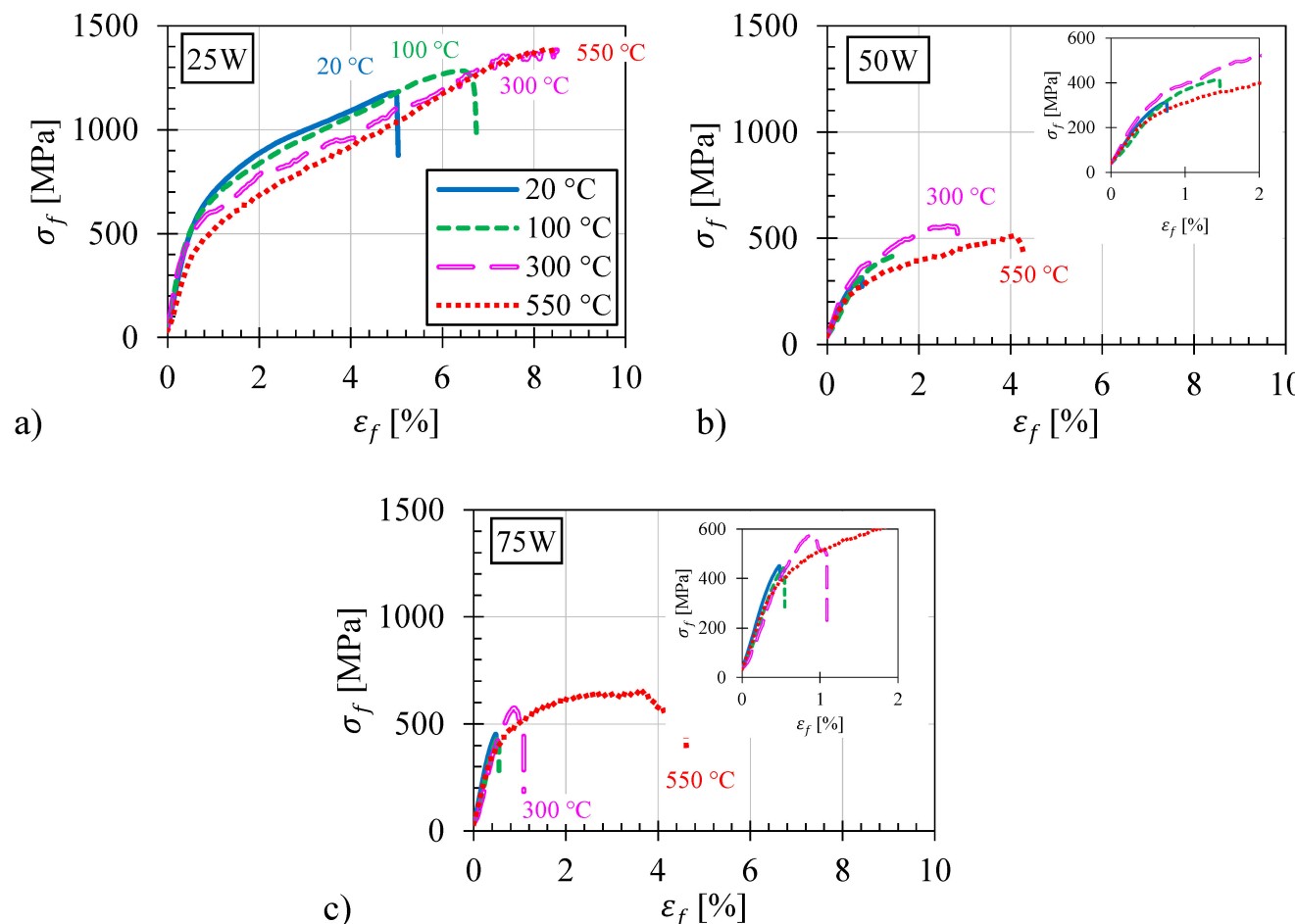

**Figure 8.** Flexural stress vs strain curves for: (**a**) 25 W; (**b**) 50 W; (**c**) 75 W composite.

### 3.4.3. Thermophysical Properties

The density measured by the Archimedes' principle is shown in Figure 9. The relative density of the composites shows that the composites achieved a promising high density, even the 75 W with a relative density of 96%. The measured (mea.) secant CTE and specific heat capacity of the composites show a good agreement with the theoretical (th.) values, as shown in Figure 10. The CTE of the composites gradually change from 25 W to 75 W and the values are in between that of pure W and pure Eurofer 97 steel. The $c_p$ curve shows a clear Curie transition peak at around 750 °C for all composites. Additionally, the intensity of the peak matches with the amount of steel present in the specimen.

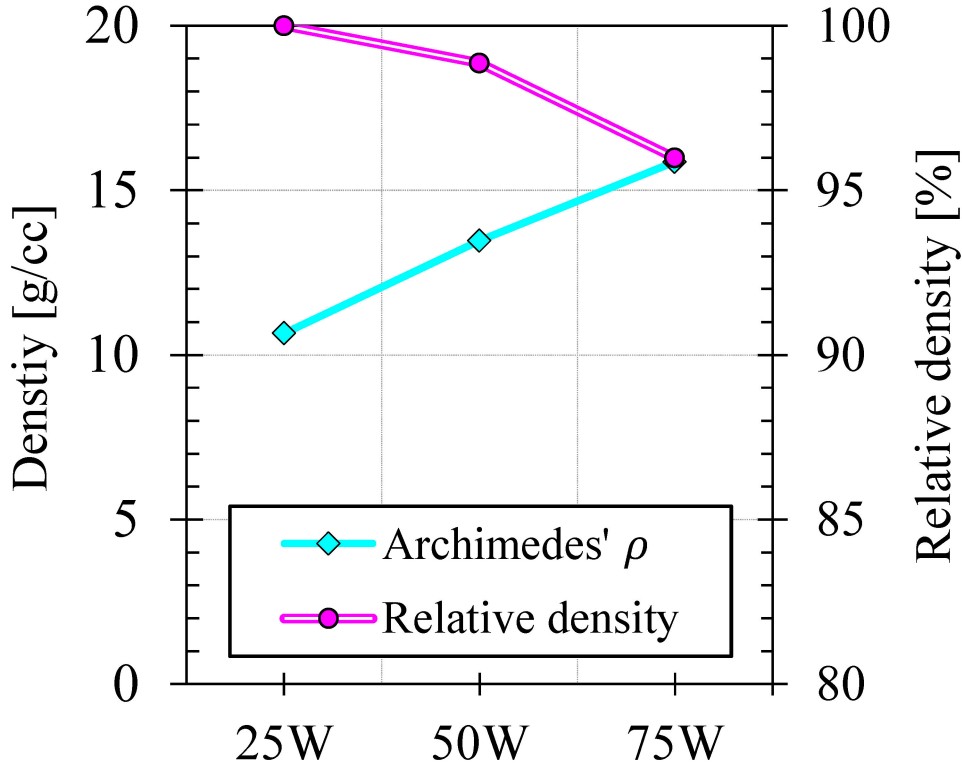

**Figure 9.** Archimedes' density and the corresponding relative density.

The measured thermal conductivities of the composites are more inclined towards the theoretical lower bound values or even below, as shown in Figure 11. For the 25 W and 50 W composites, despite having a relative density of 99%, the thermal conductivities follow the lower bound value. This is because of the spatial arrangement of the W particles. The W particles are mostly surrounded by the steel matrix, which is analogous to a metal matrix composite, with W particles being embedded inside the steel matrix. This implies that the heat must pass the steel and W constituents successively (one after the other). Equation (4), which predicts the lower bound values for composites where the heat follows the constituents successively, is therefore more suitable for 25 W and 50 W composites. In the case of 75 W, the temperature dependency of the thermal conductivity follows the theoretical upper bound model (Equation (3)). Here, thermal conduction is dominated by the W contribution. However, the absolute values are much lower than the predicted ones. This implies a large thermal resistance within the material due to the weak or even non-existing bonding between W particles.

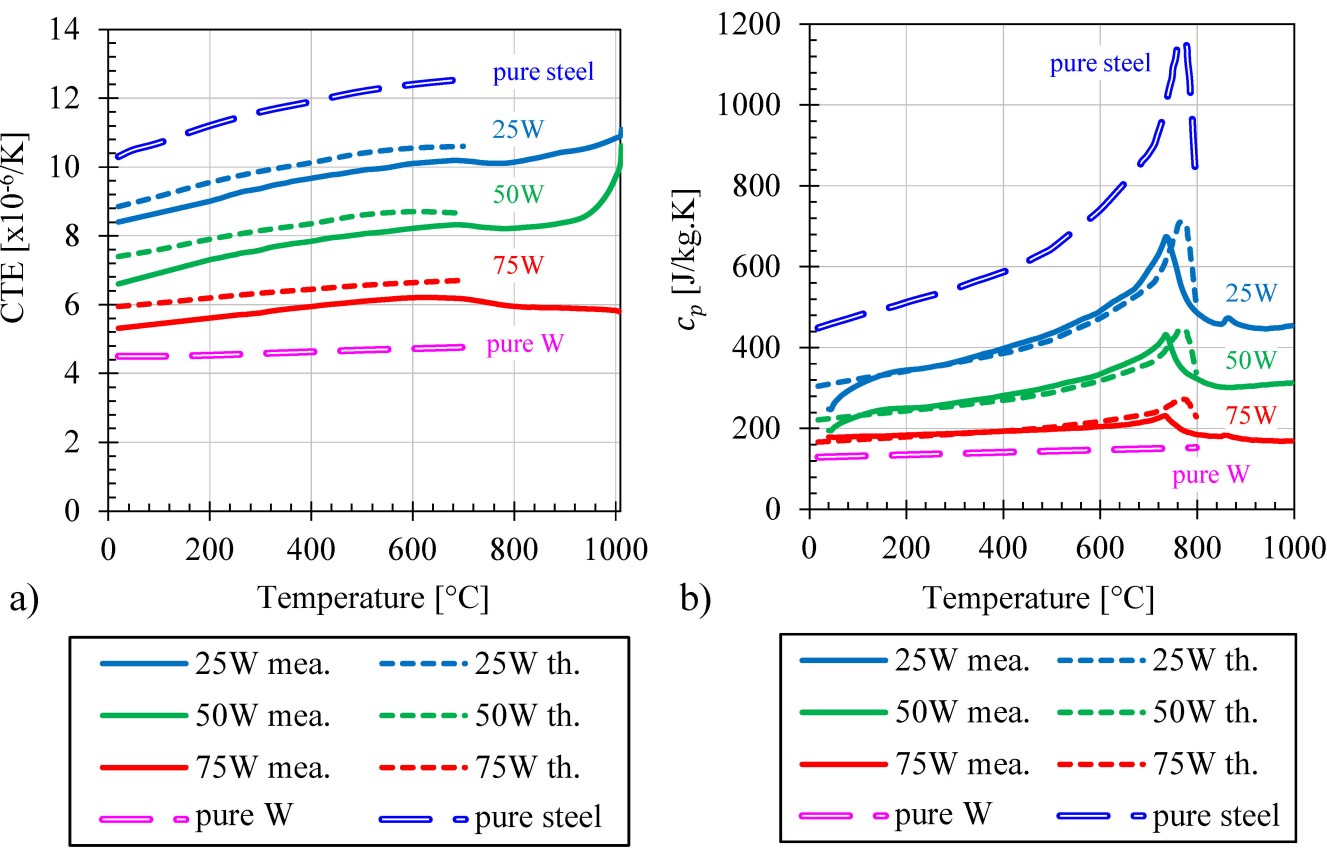

**Figure 10.** (**a**) Secant CTE; (**b**) Specific heat capacity of the composites in comparison to their theoretical values. In addition, values of pure W and pure Eurofer 97 steel [24–26].

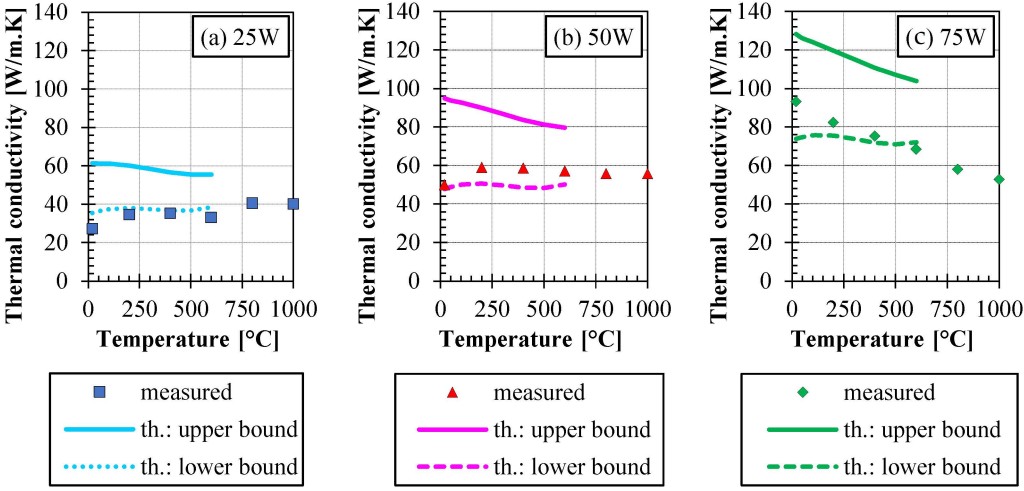

**Figure 11.** Measured and theoretically expected lower and upper bound values: (**a**) 25 W; (**b**) 50 W; and (**c**) 75 W composite.

## 4. Conclusions

In the present work, FAST/SPS technique was used to manufacture W/steel composites with three W concentrations: 25, 50 and 75 vol% W (25 W, 50 W and 75 W). These composites will be used as a stress-relieving functionally graded (FGM) interlayer for W–steel joining. Optimum sintering parameters were found in this comprehensive study to be 1000 °C, 125 MPa, and 5 min. The optimized 25 W and 50 W composites were successfully manufactured with less than 1% porosity. Using the same parameters also for 75 W resulted in a relative density of 96%. The optimized composites also contained a

low amount of brittle intermetallic compounds. Microstructural analysis of the composites revealed the presence of martensite and ferrite phases in the steel matrix. The thermophysical properties of the composites agree very well with theoretical values calculated on the basis of simple rules of mixtures. The CTE of the composites gradually varies according to the volume concentration of W, which means that these composites are suitable to be used as sublayers of the FGM. The thermal conductivities of the composites agreed well with the expected theoretical lower bound values and were higher than that of pure Eurofer 97 steel. However, despite the high relative density of the 50 W and 75 W composites, their flexural mechanical properties were worse than that of the 25 W composite. Nevertheless, all composites still showed a reasonable ductility above 300 °C; it must be pointed out that the temperature of the coolant in the first wall will be about 300 °C, and this means the lowest temperature such composites would undergo during operation is 300 °C. This further implies that such composites would withstand the thermal stresses occurring during the thermal cycling in a future fusion reactor.

In addition, a brief comparison of the composites produced in this work with other previous works shows the effectiveness of our approach. Firstly, the composites produced in this work had a higher density than equivalent composites from the literature. In the work of Koller et al. [17], 43 W and 69 W composites prepared by FAST/SPS at 1100 °C had a lower relative density of 95% and 82%. The 50 W and 75 W composites produced in this work had a relative density of 99% and 96%, respectively. Secondly, the optimized sintering parameters limited the amount of IMC successfully. For comparison, the same 43 W composite prepared by Koller et al. [17] contained around 12% IMC, while the 50 W composite produced in this work contained only 5% IMC. Thirdly, the thickness of the IMC belt formed at the W–steel interface was also drastically reduced. For comparison, a 50 W composite prepared by FAST/SPS at 1050 °C by Tan et al. [16] contained an IMC belt with a thickness of 2 μm. In this work, the IMC belt thickness in a 50 W composite was only ~200 nm. Fourthly, the composites prepared in this work have better mechanical properties. For comparison, a 25 W composite prepared by FAST/SPS at 1100 °C by Matejicek et al. [29] had a low flexural strength of 350 MPa when tested at room temperature, and the fractured surface of the samples indicated a brittle failure. The 25 W composite produced in this work was ductile at room temperature and had a high flexural strength of 1150 MPa when tested at room temperature. Although this brief comparison shows that the composites prepared in this work have better properties compared to other studies, there is still room for further improvements. Especially for the 75 W composites, achieving a density beyond 96% would be the key with regard to this. Better densification can be achieved e.g., by the use of high-pressure tools made of Mo-based TZM material instead of graphite, enabling pressure up to ~400 MPa.

As an intermediate step, a complete graded FGM was also sintered by placing the 25 W, 50 W and 75 W premixed powders on top of each other and sintering at the optimized sintering parameter. A cross-sectional overview is provided in Figure S8 in Supplementary Materials. Furthermore, the gained knowledge and properties of the composites would help to understand the potential application of such an FGM interlayer.

**Supplementary Materials:** The following supporting information can be downloaded at: https://www.mdpi.com/article/10.3390/jne4010014/s1, Figure S1: SEM images of some of the mixed W-steel powders; Figure S2: Technical drawing of the optimized punch and die made of graphite; Figure S3 to Figure S5: Additional SEM micrographs of sintered 25 W composites; Figure S6: Additional SEM micrographs of sintered 50 W composites; Figure S7: Additional SEM micrographs of sintered 75 W composites; Figure S8: Cross-sectional micrograph of the manufactured FGM.

**Author Contributions:** Conceptualization, V.G. and D.D.-G.; methodology, V.G. and D.D.-G.; investigation, V.G.; resources, M.B.; writing—original draft preparation, V.G.; writing—review and editing, V.G., D.D.-G. and M.B.; supervision, D.D.-G. and W.T.; project administration, C.L.; funding acquisition, J.W.C., M.W. and G.P. All authors have read and agreed to the published version of the manuscript.

**Funding:** This work was carried out within the framework of the EUROfusion Consortium, funded by the European Union via the Euratom Research and Training Programme (Grant Agreement No 101052200–EUROfusion). Views and opinions expressed are, however, those of the authors only and do not necessarily reflect those of the European Union or the European Commission. Neither the European Union nor the European Commission can be held responsible for them.

**Acknowledgments:** The authors would like to thank the following people for their assistance: Beatrix Göths for FIB cuts, Rudi Caspers for specimen cutting, Philipp Lied and Siegfried Baumgärtner for bending tests.

**Conflicts of Interest:** The authors declare no conflict of interest. The funders had no role in the design of the study; in the collection, analyses, or interpretation of data; in the writing of the manuscript; or in the decision to publish the results.

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
