# Peer review of "Processing and Properties of Sintered W/Steel Composites for the First Wall of Future Fusion Reactor"

_jne, doi:10.3390/jne4010014_

Round 1
Reviewer 1 Report
Nowadays the first wall material for a fusion reactor is becoming more and more important. This manuscript has shown one way to use the metal sintering technique to form the composites of W and Steel. 25%,50%,75%W with steels have been tested, the CTE, the specific heat capacity, stress, thermal conductivity have been tested and compared, optimized, the results are exciting and interesting, The optimized 25W and 50W composites were successfully manufactured with less than 1 % porosity. Using the same parameters also for 75W resulted in a relative density of 96 %.Optimum sintering parameters have been found in this comprehensive study to be 1000 °C 125 MPa, and 5 min. all composites still showed a reasonable ductility above 300 °C.
Author Response
We thank the reviewer for the positive rating of our work and for taking the time to read.
Reviewer 2 Report
Review Report
The manuscript titled” Processing and Properties of sintered W/ steel composites for the I wall of future fusion reactor” addresses mainly the mechanical and thermo physical properties of W/steel composite prepared using Spark Plasma Sintering. The motivation of the work is relevant with respect to application in the first wall of fusion reactor and welcome.
However the authors appears to have not followed the correct methodology in carrying out this work in terms of the sample preparation as would be made clear based on the following questions arising out of this work.
1) Authors have mentioned the ratio of the composition by means of volume which may not be a correct way of quantifying while these two constituents steel and tungsten have widely different values of densities
2) This work suffers from serious mistake in the methodology of preparation of the composite by means of tumble mixing . High energy ball milling could have been the far superior choice of preparing the composite. Therefore in the present case , could you explain as to how the chemical homogeneity of the particles of steel and tungsten are ascertained.
3) Another serious flaw in the sample preparation would be due to different sizes of W and steel particles used for preparing the composite. This may have a large impact on densification characterization, homogenization which could have serious implications on the microstructure , mechanical and thermo physical properties. Authors should give a serious consideration on these points.
4) Difference in particle sizes could be the major reason for the difference in porosity of the considered product of different thickness rather than wall size effect as claimed by the Authors
5) Based on the above points the optimization condition as mentioned in this work could not be true representation of the system as claimed in the paper.
6) How did you evaluate the relative density of different compositions. How do you comprehend the changes in the relative density as a function of composition
Summarizing , though the motivation of this work is good , the methodology adopted to obtain the required system seems to be seriously incorrect. This would therefore have a serious implications on the observed mechanical and thermo physical properties.
Hence it is not possible to recommend this work for a possible publication in the journal of Nuclear Engineering.
Author Response
"Please see the attachment"

Reviewer 3 Report
The authors analysed a new sintering technique to produce steel-W composites at different W percentages (25%, 50% and 75%).
In nuclear fusion reactors, the plasma-facing components will be made of tungsten while the structural material will be made of steel. Therefore, the production of these composites is relevant to be used as interlayers, aiming at minimising the stress arising from the different thermal expansions of the two materials.
In this paper, the authors analysed the composite properties as a function of the sintering parameters (mainly temperature and pressure) and the properties of the W and steel powders. The authors provided an analysis of the porosity of the composites, their mechanical characterisation (stress-strain curve), thermophysical characterisation (CTE, specific heat capacity, thermal conductivity). The results are compared with physical/theoretical expectations.
The paper is well presented. It is divided into introduction, material and methods, results and discussions, and conclusions. The subsections are well organised too, allowing easy reading and understanding of the paper.
My unique double is about the introduction, which is quite short and does not provide insight respect with to other techniques. Are there other methods to produce these composites? if yes, what are the advantages and disadvantages of using this new methodology? Probably a brief section discussing other alternatives may be important.
However, in my opinion, this is just a minor comment that does not affect my general good opinion of the paper, which may be published as it is. Therefore, in the final decision, I just selected "minor comments" and I leave to the editor the choice to ask for an introduction improvement.
Just another minor comment. At the beginning of section 2.1, you speak about D50. It may useful to briefly explain what this parameter represents since the journal is not a material specialised journal and the readers may be not aware of the meaning.
Author Response
"Please see the attachment"
